# Checkpoint inhibition of origin firing prevents inappropriate replication outside of S-phase

**Mark C Johnson, Geylani Can, Miguel Monteiro Santos, Diana Alexander, Philip Zegerman***

Wellcome Trust/Cancer Research United Kingdom Gurdon Institute and Department of Biochemistry, University of Cambridge, Cambridge, United Kingdom

**Abstract** Checkpoints maintain the order of cell cycle events during DNA damage or incomplete replication. How the checkpoint response is tailored to different phases of the cell cycle remains poorly understood. The S-phase checkpoint for example results in the slowing of replication, which in budding yeast occurs by Rad53-dependent inhibition of the initiation factors Sld3 and Dbf4. Despite this, we show here that Rad53 phosphorylates both of these substrates throughout the cell cycle at the same sites as in S-phase, suggesting roles for this pathway beyond S-phase. Indeed, we show that Rad53-dependent inhibition of Sld3 and Dbf4 limits re-replication in G2/M, preventing gene amplification. In addition, we show that inhibition of Sld3 and Dbf4 in G1 prevents premature initiation at all origins at the G1/S transition. This study redefines the scope of the 'S-phase checkpoint' with implications for understanding checkpoint function in cancers that lack cell cycle controls.

**\*For correspondence:**
paz20@cam.ac.uk

**Competing interests:** The authors declare that no competing interests exist.

## Introduction

It is vitally important that in every cell division, the entire genome is replicated once and only once. In eukaryotes, this is achieved by linking DNA replication control to the cell cycle (*Siddiqui et al., 2013*). The first step in replication is the formation of the pre-replicative complex (pre-RC) at origins – a process called 'licensing'. Licensing involves the origin recognition complex 1-6 (Orc1-6) and Cdc6-dependent loading of double hexamers of the Mcm2–7 helicase on double-stranded DNA. Licensing is restricted to late mitosis/early G1 phase by the activity of the APC/C (anaphase-promoting complex/cyclosome), which eliminates licensing inhibitors such as cyclin-dependent kinase (CDK) and geminin in this window of the cell cycle. In the budding yeast *Saccharomyces cerevisiae*, which lacks geminin, CDK inhibits licensing from late G1 phase until mitosis by multiple mechanisms, including direct phosphorylation of Orc2/Orc6 and nuclear exclusion of the Mcm2–7 complex, and by mediating SCF[CDC4]-dependent degradation of Cdc6 (*Blow and Dutta, 2005*; *Nguyen et al., 2001*).

Importantly, Mcm2–7 double hexamers loaded in late M/early G1 phase are inactive, and replication initiation can only occur after the inactivation of the APC/C at the G1/S transition. APC/C inactivation allows the accumulation of S-phase CDK and Dbf4-dependent kinase (DDK) activities (*Labib, 2010*). DDK directly phosphorylates the inactive Mcm2–7 double hexamers, generating a binding site for firing factors including Sld3/Sld7 and Cdc45, while CDK phosphorylates Sld3 and an additional initiation factor Sld2, which via phospho-interactions with Dpb11, results in replisome assembly by poorly understood mechanisms (*Riera et al., 2017*). This duality of function of CDK, both as an inhibitor of licensing and as an activator of the replisome, is critical to ensure once per cell cycle replication (*Diffley, 2004*).

In light of the importance of the linkage between DNA replication control and cell cycle progression, multiple checkpoints exist to regulate DNA synthesis and genome integrity before (G1 checkpoint), during (S-phase checkpoint), and after S-phase (G2/M checkpoint, *Hartwell and Weinert, 1989*; *Kastan and Bartek, 2004*). These checkpoints are mediated by the PI3 kinase superfamily checkpoint kinases ATM/ATR (Ataxia telangiectasia mutated/*ATM* and RAD3-related, called Tel1/Mec1 in budding yeast) and the effector checkpoint kinases Chk1/Chk2 (Chk1/Rad53 in budding yeast).

In G1 phase, DNA damage such as UV photoproducts causes checkpoint-dependent delays in the onset of DNA replication by inhibition of G1/S cyclin–CDK activity (*Lanz et al., 2019*; *Shaltiel et al., 2015*). In humans, the G1–S transition is delayed by ATM-Chk2-mediated stabilisation of p53, resulting in expression of the CDK inhibitor p21, as well as by checkpoint-dependent degradation of cyclin D and the CDK activator Cdc25A (*Lanz et al., 2019*; *Shaltiel et al., 2015*). In budding yeast, the delay in G1–S occurs in part by Rad53-dependent phosphorylation and inhibition of the Swi6 subunit of the transcriptional activator SBF (SCB binding factor) leading to reduced cyclin transcription (*Sidorova and Breeden, 1997*).

In S-phase, origin firing in unperturbed cells occurs as a continuum throughout S-phase, with some origins firing in the first half (early origins) or the second half of S-phase (late origins). When replication forks emanating from early firing origins stall, for example due to DNA lesions, activation of the S-phase checkpoint kinase response results in the dramatic slowing of replication rates (*Painter and Young, 1980*; *Paulovich and Hartwell, 1995*), which occurs in large part through inhibition of late firing origins (*Yekezare et al., 2013*).

Similar to G1 phase, checkpoint activation in S-phase in human cells can prevent origin firing through inhibition of CDK activity, for example by Chk1- and Chk2-mediated degradation of the CDK activator Cdc25A (*Bartek et al., 2004*). In budding yeast however, the checkpoint kinase Rad53 blocks late origin firing, not by inhibition of CDK activity, but by directly inhibiting two replication initiation factors: the DDK subunit Dbf4 and the CDK target Sld3 (*Lopez-Mosqueda et al., 2010*; *Zegerman and Diffley, 2010*). Rad53-dependent phosphorylation of Sld3 inhibits its interactions with various other replication factors including Dpb11 (*Lopez-Mosqueda et al., 2010*; *Zegerman and Diffley, 2010*), while recent work has shown that direct interaction between Rad53 and Cdc7-Dbf4 prevents DDK from binding to the Mcm2–7 complex (*Abd Wahab and Remus, 2020*). In higher eukaryotes, in addition to inhibition of CDK activity, the checkpoint can also inhibit origin firing through direct targeting of the Sld3 orthologue Treslin (*Boos et al., 2011*; *Guo et al., 2015*) and DDK (*Costanzo et al., 2003*; *Lee et al., 2012*). Therefore, multiple mechanisms exist to ensure the robust inhibition of replication initiation in S-phase by direct targeting of Sld3/Treslin and Dbf4-dependent kinase and by inhibition of CDK activity (*Lanz et al., 2019*). One function of such robust inhibition of origin firing during S-phase in the presence of DNA lesions is to prevent the exhaustion of essential factors, such as topoisomerase activities, by excessive numbers of replisomes (*Morafraile et al., 2019*; *Toledo et al., 2017*).

A key proposed feature of the DNA damage checkpoints is that the response is tailored to the cell cycle phase in which the DNA damage occurred (*Shaltiel et al., 2015*). Despite this, there is very little evidence to suggest that substrate specificity of the checkpoint kinases changes during the cell cycle. Indeed, in budding yeast, most forms of DNA damage and replication stress converge on the single effector kinase Rad53, but how different checkpoint responses in different cell cycle phases can be mediated by a single kinase is not known. In this study, we set out to explore the specificity of Rad53 towards the replication substrates Sld3 and Dbf4 across the cell cycle in the budding yeast, *S. cerevisiae*. We show that Rad53 phosphorylates both of these substrates throughout the cell cycle at the same sites as in S-phase. From this we hypothesised that although these substrates are deemed to be targets of the 'S-phase checkpoint', Rad53 may also prevent aberrant origin firing outside of S-phase. Indeed, we show that Rad53-dependent inhibition of Sld3 and Dbf4 limits re-initiation of replication in G2/M phase and also prevents premature firing of all origins, not just late origins, at the G1/S transition. This study overhauls our understanding of the cell cycle phase specificity of the 'S-phase checkpoint' and provides a novel mechanism that restricts replication initiation to a specific window of the cell cycle after DNA damage.

## Results and discussion

### Sld3 and Dbf4 are phosphorylated by Rad53 outside of S-phase

Since the DNA damage checkpoint response can be activated in all phases of the cell cycle, we addressed whether the replication factors Sld3 and Dbf4 could be targeted by Rad53 outside of S-phase *in vivo* in budding yeast. To test this, we first analysed the consequences of DNA damage in G1 phase cells arrested with the mating pheromone alpha factor (*Figure 1A*). These experiments were conducted with strains containing a null mutation in the alpha factor protease, *bar1Δ*, to ensure that cells were fully arrested in G1 phase and had not started DNA replication. Addition of the UV mimetic drug 4-NQO to G1 phase cells resulted in robust Rad53 activation, as determined by the accumulation of the phospho-shifted forms of the kinase (*Figure 1B,C*). Importantly, we observed a dramatic increase in lower mobility forms of Sld3 when Rad53 was activated in G1 phase (*Figure 1B*), which was indeed Rad53 dependent (*Figure 1—figure supplement 1B*). In line with a previous study (*Mattarocci et al., 2014*), in G1-arrested cells in the absence of 4-NQO (*Figure 1B*) or in *rad53Δ* cells (*Figure 1—figure supplement 1B*), Sld3 can be seen to migrate as a doublet on phos-tag gels. This was shown to be caused by DDK phosphorylation of Sld3 (*Mattarocci et al., 2014*), but does not affect the DNA damage and Rad53-dependent phosphorylation of Sld3 described here (*Figure 1B*). For Dbf4, which is an APC/C substrate and partially degraded in alpha factor-arrested cells (*Ferreira et al., 2000*), we also observed a mobility shift in G1 phase coincident with Rad53 activation (*Figure 1C*).

To test whether Sld3 and Dbf4 could also be phosphorylated by Rad53 after DNA replication is complete, we performed the same experiment as in *Figure 1A–C*, except in cells arrested in G2/M phase with nocodazole (*Figure 1D*). Significantly, we observed a Rad53-dependent mobility shift in Sld3 and Dbf4, even in G2/M-arrested cells (*Figure 1E,F*). Sld3 and Dbf4 are phosphorylated by other kinases in G2/M, such as by CDK (*Holt et al., 2009*), giving rise to additional isoforms of Sld3/Dbf4 proteins even in Rad53 null cells (* *Figure 1E,F*). Note that the CDK-phosphorylated form of Sld3 is visible in *Figure 1E* as this is a phos-tag gel, and we confirmed that both of the bands in *rad53Δ* cells in *Figure 1E* are Sld3 dependent and the upper form is indeed CDK dependent (*Figure 1—figure supplement 1C*).

Previously, we have identified the serine and threonine residues in Sld3 and Dbf4 that are directly phosphorylated by Rad53 in S-phase (*Zegerman and Diffley, 2010*). We mapped 38 such phospho-sites in Sld3 and 19 sites in Dbf4. Mutation of these serine/threonine residues to alanine in Sld3- and Dbf4-generated alleles that are refractory to Rad53 phosphorylation in S-phase (*Zegerman and Diffley, 2010*) and are hereafter referred to as *sld3-A* and *dbf4-19A*, respectively. We reasoned that if the same sites in Sld3 and Dbf4 are phosphorylated by Rad53 throughout the cell cycle, then *sld3-A* and *dbf4-19A* should be defective in Rad53-dependent phosphorylation in G1 and G2/M as well. In G1 phase, the Sld3-A protein demonstrated a dramatic loss of Rad53 phosphorylation (*Figure 2A*), consistent with direct phosphorylation of Sld3 by Rad53 at the same sites as in S-phase. We also observed a similar result with the Dbf4-19A protein in G1 phase after Rad53 activation (*Figure 2B*). As in G1 phase, both Sld3-A and Dbf4-19A showed greatly reduced phosphorylation during Rad53 activation in G2/M phase (*Figure 2C,D*). Together, *Figures 1* and *2* show that although Sld3 and Dbf4 are considered to be 'S-phase checkpoint' substrates of Rad53, they are phosphorylated at the same sites as in S-phase after DNA damage in G1 and G2 phase.

The *dbf4-19A* mutant, although refractory to Rad53 phosphorylation, is also a hypomorph and is not fully functional for replication initiation (*Zegerman and Diffley, 2010*). We previously narrowed down the critical residues that are inhibited by Rad53 to four sites and generated a *dbf4-4A* allele, which still cannot be inhibited by Rad53, but is fully functional for its role in replication initiation (*Zegerman and Diffley, 2010*). In similarity to the 19A allele, Dbf4-4A was less Rad53-phosphorylated than wild type in G1 cells after DNA damage (*Figure 2—figure supplement 1*), suggesting that it is indeed the critical residues in Dbf4 that are phosphorylated by Rad53 outside of S-phase. As this 4A allele is fully functional for Dbf4's role in replication initiation, from here on we only use this allele and refer to it as *dbf4-A*.

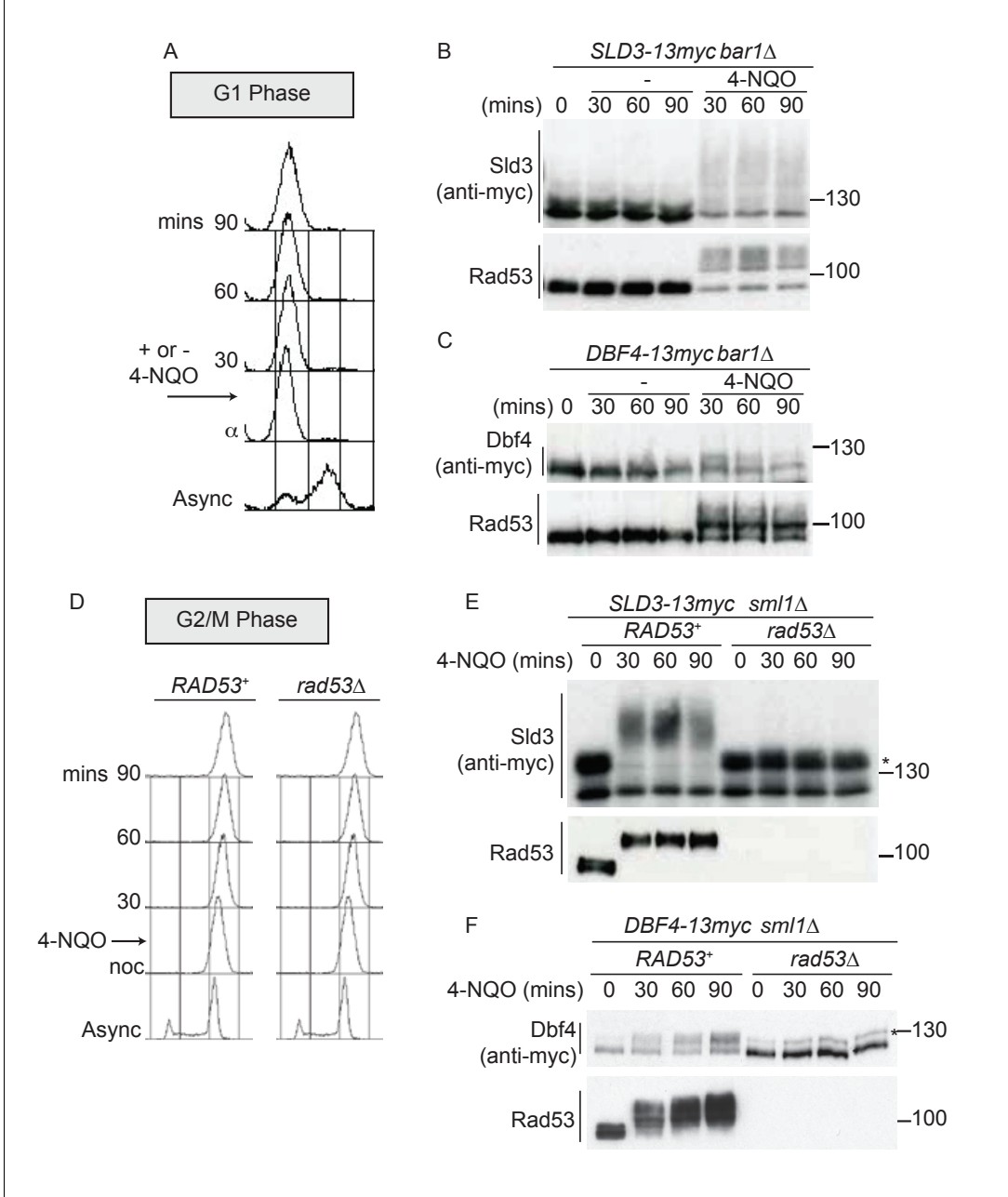

**Figure 1.** Dbf4 and Sld3 are phosphorylated by Rad53 after DNA damage in G1 and G2 phase. (A) Flow cytometry of strains arrested in G1 phase with the mating pheromone alpha factor. Strains were held in G1 phase, with or without the addition of 10 μg/ml 4-NQO for the indicated times. All strains are *bar1Δ* to maintain G1 arrest. (B) Western blot of Sld3 (anti-myc) and Rad53 phosphorylation from the experiment outlined in (A). Sld3 was resolved on a phos-tag sodium dodecyl sulfate–polyacrylamide gel electrophoresis (SDS–PAGE) gel. (C) As (B), but for Dbf4. Both blots are from SDS–PAGE. (D) As (A), except strains were arrested in G2/M with nocodazole before the addition of 4-NQO. All strains are *sml1Δ*, which is required for viability in cells lacking Rad53. (E) Western blot of Sld3 (anti-myc) and Rad53 phosphorylation as in (B) from the experiment outlined in (D). Sld3 was resolved on a phos-tag SDS–PAGE gel. *Cyclin-dependent kinase (CDK)-phosphorylated Sld3 (see *Figure 1—figure supplement 1C*). (F) As (E), but for Dbf4. Dbf4 is phosphorylated by other kinases in G2/M, resulting in residual phosphorylated forms remaining in *rad53Δ* cells *.

The online version of this article includes the following figure supplement(s) for figure 1:

**Figure supplement 1.** Sld3 phosphorylation in G1 phase after DNA damage is Rad53 dependent.

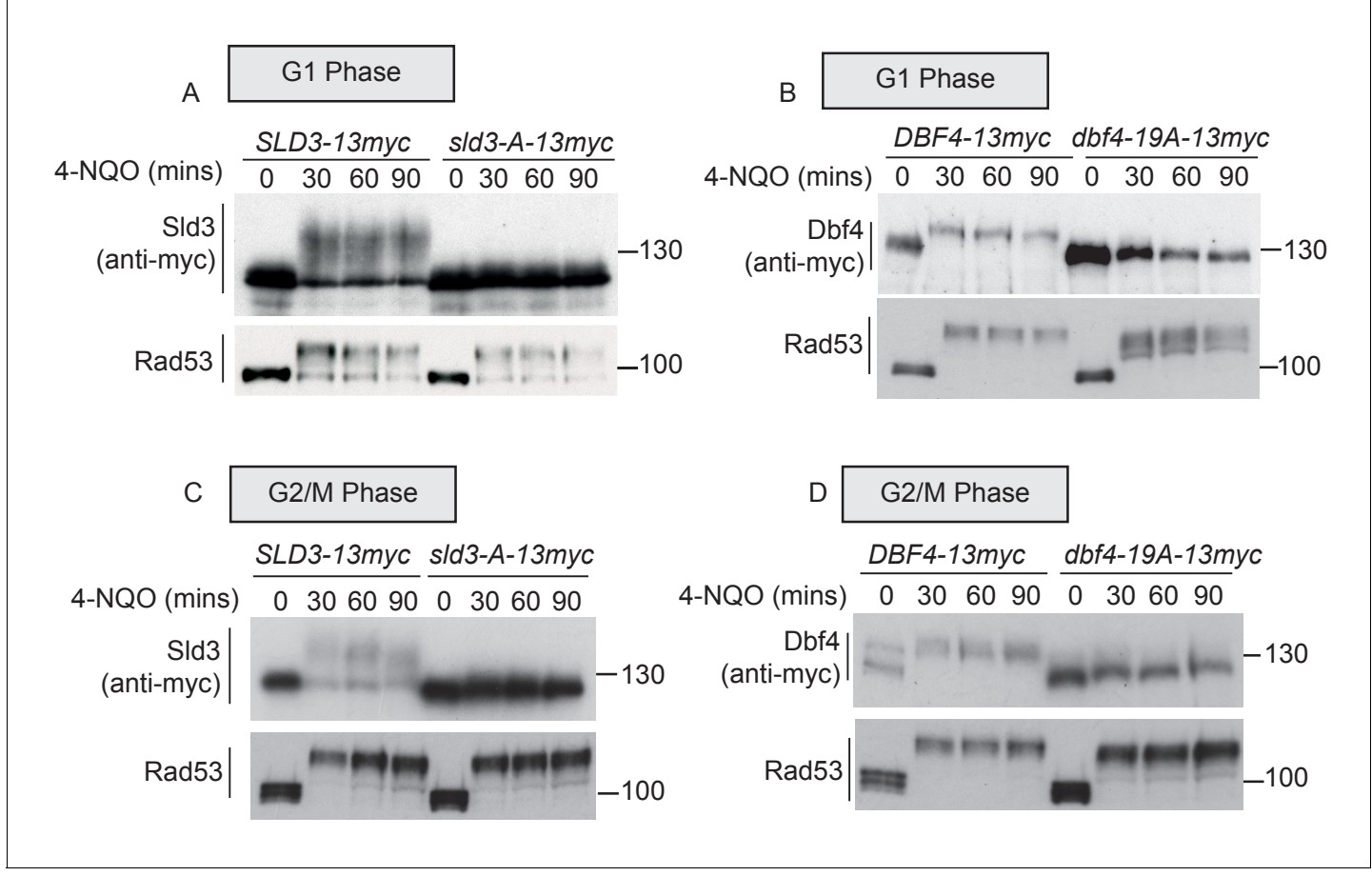

**Figure 2.** Rad53 phosphorylates Sld3 and Dbf4 in G1 and G2 phases at the same residues as in S-phase (A and B) as *Figure 1B,C*. *sld3-A* and *dbf4-19A* refer to mutant alleles with Rad53 phosphorylation sites mutated to alanine (38 sites for Sld3 and 19 sites for Dbf4). (C) As *Figure 1E*, except this is western blot from an SDS–PAGE gel. (D) As *Figure 1F*.

The online version of this article includes the following figure supplement(s) for figure 2:

**Figure supplement 1.** Dbf4-4A is expressed at similar levels to wild type and is also refractory to Rad53 phosphorylation in G1 phase.

## Rad53-dependent phosphorylation of Sld3 and Dbf4 reduces re-replication in G2 phase

We have previously shown that Rad53 phosphorylates Sld3 and Dbf4 in S-phase to inhibit origin firing (*Lopez-Mosqueda et al., 2010*; *Zegerman and Diffley, 2010*). As Sld3 and Dbf4 are phosphorylated at the same sites by Rad53 after DNA damage in both G1 and G2 phase (*Figures 1* and *2*), we wondered whether this phosphorylation could also be required to inhibit replication initiation outside of S-phase. DNA replication is tightly restricted to S-phase in large part by the action of CDK, which prevents licensing outside of late M/early G1 phase. As a result, transient reduction of CDK activity in G2/M phase is sufficient to induce re-replication (*Dahmann et al., 1995*). To test a role for Rad53 phosphorylation of Sld3/Dbf4 in re-replication control, we first combined the *sld3-A/ dbf4-A* alleles, which cannot be inhibited by Rad53 (*Zegerman and Diffley, 2010*), with a hypomorphic mutant of the CDK catalytic subunit Cdc28 (*cdc28-as1*). This allele of Cdc28 is analogue sensitive and is inhibited by the addition of the ATP-competitive inhibitor 1-NM-PP1. Interestingly, we observed that the *sld3-A/dbf4-A* alleles are synthetically sick with *cdc28-as1* in the presence of sublethal doses of 1-NM-PP1 (*Figure 3—figure supplement 1A*), suggesting that inhibition of Sld3 and Dbf4 by Rad53 is important in cells that have reduced CDK activity.

To specifically test whether the Rad53-dependent inhibition of origin firing is important to prevent re-replication, we combined the *sld3-A/dbf4-A* alleles with mutants that circumvent the CDK-dependent inhibition of licensing. Over-expression of Cdc6, forcing the nuclear localisation of the

Mcm2-7 complex (through an Mcm7-2xNLS fusion) and mutation of the CDK phosphorylation sites in ORC is sufficient to induce re-replication in G2/M phase (*Finn and Li, 2013*; *Nguyen et al., 2001*) and has been shown to induce Rad53 activation (*Archambault et al., 2005*; *Green and Li, 2005*). Importantly, conditional over-expression of licensing mutants that cannot be inhibited by CDK combined with *sld3-A* and *dbf4-A* led to an increase in the total re-replication in nocodazole arrested cells (*Figure 3A* – compare flow cytometry overlay red vs. black).

To analyse whether this shift in DNA content by flow cytometry was indeed due to genomic re-replication, we performed DNA sequencing and copy number analysis of the experiment in *Figure 3A*. While we did not detect any differences between the mitochondrial DNA content of the strains (*Figure 3—figure supplement 1B*), we observed a dramatic increase in DNA copy number largely for Chromosome III in strains that over-express Cdc6 and pre-RC mutants that cannot be inhibited by CDK (*Figure 3B*, *Figure 3—figure supplement 1C*). Previous studies have indeed shown that chromosome III is particularly susceptible to re-replication (*Green et al., 2006*). While we did not detect any re-replication in the *sld3-A/dbf4-A* strain alone (data not shown), we observed greater re-replication of chromosome III when the pre-RC mutants were combined with *sld3-A* and *dbf4-A*, which cannot be inhibited by Rad53 (*Figure 3B*). Together with *Figure 3A*, this suggests that Rad53-dependent inhibition of replication initiation can reduce inappropriate replication in G2 phase when licensing control is compromised.

One of the consequences of re-replication is the generation of head-to-tail tandem gene amplifications, a process termed re-replication-induced gene amplification (RRIGA) (*Green et al., 2010*). To examine whether the Rad53-dependent inhibition of origin firing helps to prevent RRIGA we adapted an assay to quantitatively assess gene amplification events in G2/M arrested cells (*Finn and Li, 2013*). Briefly, a marker gene (in this case LEU2, which allows growth on media lacking leucine) was split with some remaining homology across an origin that re-initiates when licensing control is lost (ARS317 – see for example *Figure 3B*). Re-initiation at ARS317 followed by fork-breakage and strand annealing at the regions of LEU2 homology results in gene amplification and the generation of a functional LEU2 gene (*Figure 3C*, *Figure 3—figure supplement 1D,E*). In this assay, as in *Figure 3A*, the re-replication mutants were induced only in G2/M-arrested cells. In contrast to wild-type yeast or the *sld3-A/dbf4-A* strain alone, expression of the licensing mutants by the addition of galactose resulted in a large increase in RRIGA events, as expected (*Figure 3D*). Importantly, RRIGA events were even greater when the mutants that allow licensing in the presence of CDK were combined with the *sld3-A/dbf4-A* alleles (*Figure 3D*). This assay demonstrates that the checkpoint kinase Rad53 indeed reduces gene amplification events after re-replication through inhibition of Sld3 and Dbf4, even in G2/M-arrested cells.

## Rad53 prevents precocious origin firing after DNA damage in G1 phase

DNA damage in G1 phase delays the G1/S transition, which from humans to yeast, involves the checkpoint kinase-dependent down-regulation and/or inhibition of G1/S cyclin-CDK activity (*Bertoli et al., 2013*; *Lanz et al., 2019*; *Shaltiel et al., 2015*; *Sidorova and Breeden, 1997*). Here we have shown that DNA damage in G1 phase also results in the inhibitory checkpoint phosphorylation of two replication initiation factors, Sld3 and Dbf4 (*Figures 1* and *2*), suggesting that this might be an additional mechanism to prevent premature DNA replication at the G1/S transition (*Figure 4A*). To specifically analyse the consequences of DNA damage in G1 phase, we added 4-NQO to G1-arrested yeast cells and then released cells into S-phase in fresh medium without 4-NQO. Crucially, this approach resulted in robust Rad53 activation in G1 phase, such that cells enter S-phase with an already active checkpoint (*Figure 4D*, *Figure 4—figure supplement 1*). Rad53 activation in G1 phase resulted in the slowing of the G1/S transition as detected by the delay in budding (a G1 cyclin/CDK-mediated event) and the delay in DNA synthesis (*Figure 4—figure supplement 1*). Despite this, the *sld3-A dbf4-A* alleles caused little difference in S-phase progression after DNA damage in G1 phase, compared to the wild-type strain (*Figure 4—figure supplement 1*).

As Rad53 is known to inhibit CDK activation through phosphorylation of the Swi6 subunit of the transcriptional activator SBF (*Sidorova and Breeden, 1997*), we wondered whether Rad53-dependent inhibition of both replication initiation factors and G1/S transcription might prevent precocious DNA replication after damage in G1 phase (*Figure 4A*). To test this, we over-expressed a truncated form of the SBF transcription factor Swi4 (Swi4-t), which lacks the C-terminus required for interaction with Swi6 and thus cannot be inhibited by Rad53 (*Sidorova and Breeden, 1997*). Over-expression

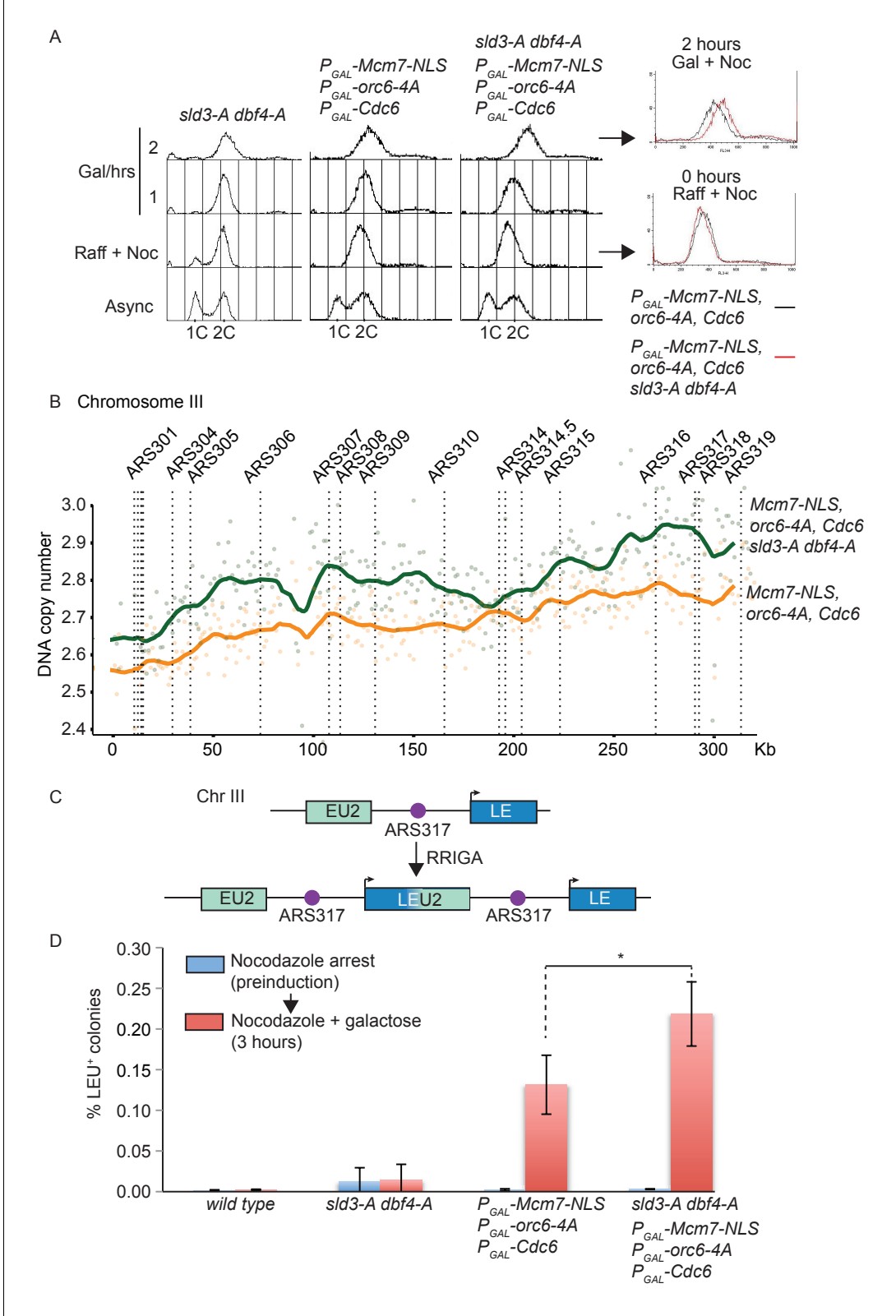

**Figure 3.** Checkpoint-dependent inhibition of origin firing prevents re-replication in G2 phase. (A) Flow cytometry of the indicated strains grown overnight in YPraffinose, then arrested in G2/M with nocodazole. After addition of fresh nocodazole, 2% galactose was added for the indicated times to express the licensing mutants. Right, overlay between the 0 and 2 hr time points for the licensing mutant strain with (red) or without (black) the *sld3-A/dbf4-A* alleles. (B) Copy number analysis of Chromosome III from the experiment in (A), after 4 hr in galactose. As in (A), the strains overexpress wild

*Figure 3 continued on next page*

*Figure 3 continued*

type Cdc6 and the indicated pre-replicative complex mutants from the galactose inducible promoter. DNA sequencing read depth per 1 kb bin was normalised to the 0 hr time point for each strain. Baseline was set at two as the strains are arrested in G2. The y-axis shows the DNA copy number after these steps. (C) Schematic diagram of re-replication-induced gene amplification (RRIGA) assay for gene amplification events. Re-replication of the split LEU2 gene from origin ARS317 results in tandem head to tail gene duplications, leading to a functional LEU2 gene. (D) RRIGA assay in (C) was performed with the indicated strains. Strains were grown overnight in YPraffinose then arrested in G2/M with nocodazole (pre-induction, blue time point). After addition of fresh nocodazole, 2% galactose was added for 3 hr to express the licensing mutants (red time point). Cells were plated on YPD (viable cell count) and SC-leu plates (Leu+ count) and the % of Leu+ colonies out of the viable cell population was plotted. N = 3, error bars are SD. *The p value was calculated as 0.0481 using an unpaired t-test.

The online version of this article includes the following figure supplement(s) for figure 3:

**Figure supplement 1.** Checkpoint-dependent inhibition of origin firing prevents re-replication in G2 phase.

of Swi4-t indeed resulted in faster progression through the G1/S transition in the presence of 4-NQO (*Figure 4—figure supplement 1D*) as expected (*Sidorova and Breeden, 1997*). Importantly, the combination of expression of Swi4-t together with *sld3-A dbf4-A* resulted in much faster S-phase progression after DNA damage in G1 phase compared to Swi4-t expression alone (*Figure 4B*). These differences in the onset of DNA replication between *swi4-t* with and without *sld3-A/dbf4-A* were not due to differences in the G1/S transition as these strains budded at the same time (*Figure 4C*) and both strains also exhibited similar levels of Rad53 activation (*Figure 4D*). Together this suggests that Rad53 activation in G1 phase prevents precocious DNA replication initiation by inhibiting not only G1/S transcription, but also Sld3 and Dbf4.

Activation of Rad53 during S-phase caused by fork stalling/DNA damage at early replicons results in inhibition of subsequent (late) origin firing, which in yeast is mediated by inhibition of Sld3 and Dbf4 (*Lopez-Mosqueda et al., 2010*; *Zegerman and Diffley, 2010*). We therefore wondered whether the accelerated S-phase we observe when we combine *swi4-t* with *sld3-A dbf4-A* is simply due to the canonical S-phase checkpoint inhibition of late origin firing or whether by activating Rad53 in G1 phase we are actually causing a delay in genome duplication from all origins. To assess this, we analysed the replication dynamics of the time course in *Figure 4B–D* by high-throughput sequencing and copy number analysis. At the earliest time point (20 min), while the *swi4-t* over-expressing strain alone had barely begun to replicate (*Figure 4E*, chromosome VII as an example), the *swi4-t sld3-A dbf4-A* strain showed peaks of replication initiation at the earliest firing origins (e. g., see *, *Figure 4E*), even though Rad53 is highly activated (*Figure 4D*). By analysis of initiation at all origins, split into quintiles according to their normal firing time, we observe much greater firing of early origins in the *swi4-t sld3-A dbf4-A* strain compared to *swi4-t* alone throughout the time course (*Figure 4F*, *Figure 4—figure supplement 2*). Over time, we also observe an increase in later firing origins in the *swi4-t sld3-A dbf4-A* strain (arrows *Figure 4F*, *Figure 4—figure supplement 2*), suggesting that the relative timing of origin firing is not affected. Together this demonstrates that activation of Rad53 and inhibition of Sld3 and Dbf4 in G1 phase contribute to the mechanism preventing the onset of DNA replication from all origins, not just late firing origins, in the presence of DNA damage (*Figure 4A*).

If the checkpoint-mediated inhibition of G1/S transcription and Sld3/Dbf4 contribute to prevent precocious S-phase entry, then we hypothesised that loss of both pathways should show synthetic lethality in the presence of DNA damage. We have previously conducted genetic interaction analysis of the *sld3-A/dbf4-A* alleles with the yeast whole genome gene knock-out collection in the presence of the DNA damaging agent phleomycin (*Morafraile et al., 2019*). Loss of function of genes that result in a delay in the G1/S transition, such as *CLN2*, *SWI4*, and *BCK2* (*Di Como et al., 1995*), improved the growth of *sld3-A/dbf4-A* in the presence of phleomycin (suppressors, *Figure 4—figure supplement 3*), whereas loss of function of genes that would result in the acceleration of G1/S, such as *WHI5* and *SIC1* (*Bertoli et al., 2013*), were synthetic sick with the *sld3-A/dbf4-A* alleles (enhancers, *Figure 4—figure supplement 3*). These genetic interactions are consistent with an important role for Rad53-dependent inhibition of origin firing in preventing precocious replication after DNA damage in G1 phase.

Here we show that the two critical targets of the S-phase checkpoint-mediated inhibition of origin firing, Sld3 and Dbf4, are actually regulated by Rad53 after DNA damage throughout the cell cycle (*Figures 1* and *2*). This has important implications for understanding the consequences of

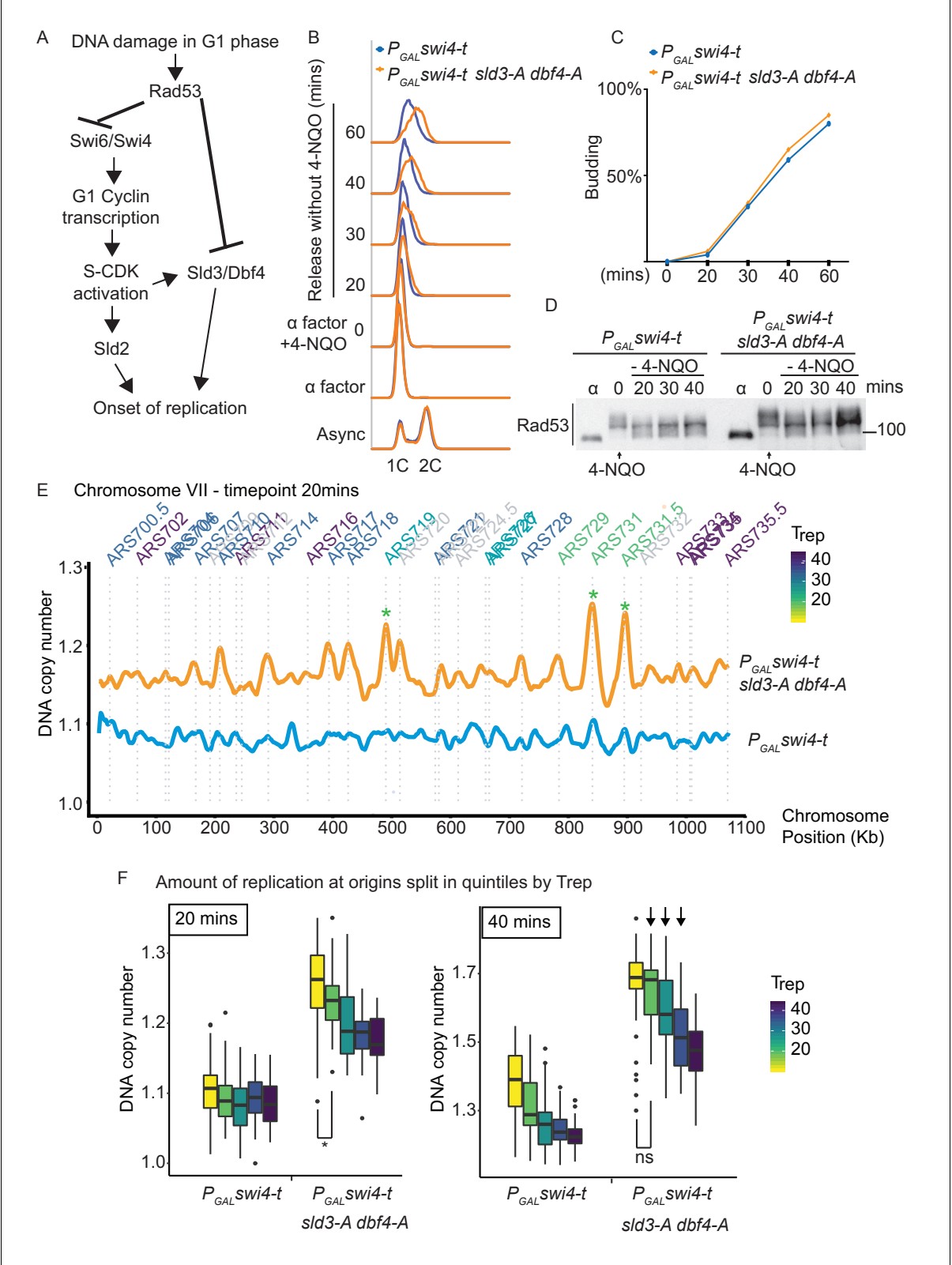

**Figure 4.** Checkpoint-dependent inhibition of origin firing prevents premature replication initiation at all origins at the G1–S transition. (**A**) Activation of Rad53 in G1 phase can delay genome duplication by at least two mechanisms; by inhibition of the transcriptional activator Swi6, which is required for G1- and subsequently S-phase cyclin transcription and by inhibition of the origin firing factors Sld3 and Dbf4. (**B**) Flow cytometry of the indicated strains grown overnight in YPraffinose, then arrested in G1 phase with alpha factor. Cells were held in fresh alpha factor, while 2% galactose and 0.5 μg/ml 4-

*Figure 4 continued*

NQO was added for 30 min (0 time point) before washing and release from alpha factor arrest into fresh YPgal medium without 4-NQO. Swi4-t is a C-terminally truncated protein that cannot be inhibited by Rad53. (C) Budding index from the experiment in (B). Time point 0 refers to cells held in alpha factor + galactose + 0.5 µg/ml 4-NQO for 30 min. (D) Rad53 western blot from experiment in (B). (E) Copy number analysis of chromosome VII of the indicated strains 20 min after release as in (B–D). The y-axis ratio refers to the amount of DNA at the 20 min time point divided by the DNA copy number in G1 phase. Known origins are annotated above the replication profile and coloured according to their normal median replication time ($T_{rep}$). (F) Box plots of the amount of replication at all origins, split into equal quintiles depending on their normal median firing time ($T_{rep}$). Initiation occurs first at early origins in the *swi4-t sld3-A dbf4-A* strain, for example the yellow and green quintiles are significantly different at 20 min (*p value 0.044), but non-significantly different (ns) at 40 min. Arrows indicate that later firing origins also initiate by 40 min in the *swi4-t sld3-A dbf4-A* strain. P-values are from t-tests.

The online version of this article includes the following figure supplement(s) for figure 4:

**Figure supplement 1.** 4-NQO addition in G1 phase delays the G1/S transition.

**Figure supplement 2.** Rad53 activation in G1 phase inhibits all origin firing through phosphorylation of Sld3 and Dbf4.

**Figure supplement 3.** Genetic interactions of the *sld3-A dbf4-A* alleles with mutants that either accelerate or delay the G1/S transition.

inappropriate re-replication in human cells, where the role of the checkpoint differs depending on the cell cycle phase in which re-replication occurs (*Klotz-Noack et al., 2012*; *Liu et al., 2007*). By combining tight cell cycle arrests with the separation-of-function mutants, *sld3-A dbf4-A*, we show specifically that the checkpoint-dependent inhibition of origin firing limits further re-replication and gene amplifications in G2 phase when licensing control is compromised (*Figure 3*). This pathway is likely to be evolutionarily conserved, as checkpoint activation in S-phase in human cells also appears to limit re-replication through inhibition of Dbf4 (*Lee et al., 2012*). As tandem head-to-tail duplications are a prominent feature of many cancers (*Menghi et al., 2018*), knowledge of the pathways that prevent this form of structural variation may be important for understanding oncogenesis.

In addition to preventing re-replication in G2, we show that the checkpoint also inhibits the replication initiation factors Sld3 and Dbf4 to delay all origin firing after DNA damage in G1 phase (*Figure 4*). This likely increases the time for DNA repair to occur before replication begins and may also serve to increase the window of time where origin firing and licensing are mutually exclusive, preventing re-replication. It is interesting that failure to inhibit Sld3 and Dbf4 alone has little effect on the G1/S transition (*Figure 4—figure supplement 1*), probably because Sld3 (and Sld2) act downstream of CDK activation (*Figure 4A*). This study therefore provides evidence that two inter-connected pathways are important for restricting replication after DNA damage in G1: inhibition of G1/S CDK activity and direct Inhibition of replication initiation factors.

Mutations in genes such as Rb and p53 that control the G1/S transition and the G1 checkpoint response, respectively, are amongst the most common mutations in cancers (*Malumbres and Barbacid, 2001*; *Massagué, 2004*). Work from yeast to humans has shown that defects in the G1/S transition results in increased dependence on the checkpoint kinases for survival (*Rundle et al., 2017*; *Sidorova and Breeden, 2002*). The identification of the checkpoint inhibition of all origin firing as a failsafe mechanism that prevents precocious S-phase entry when the G1 checkpoint is compromised described here (*Figure 4*) may provide a potential mechanistic rationale for the selective targeting of p53/Rb mutant cancers using Chk1 and ATR inhibitors, which are currently in clinical trials (*Bradbury et al., 2020*).

# Materials and methods

## Key resources table

| Reagent type (species) or resource | Designation | Source or reference | Identifiers | Additional information |
|---|---|---|---|---|
| Antibody | Anti-Rad53, rabbit polyclonal | Abcam | (ab104232) | Antibody (1:5000) |
| Antibody | Anti-c-myc, mouse monoclonal | Merck Life Science | (11667149001) | Antibody (1:1000) |
| Peptide, recombinant protein | Alpha-factor | GenScript | ( RP01002) | Peptide |

*Continued on next page*

*Continued*

| Reagent type (species) or resource | Designation | Source or reference | Identifiers | Additional information |
|---|---|---|---|---|
| Commercial assay or kit | TruSeq Nano DNA Low Throughput Library Prep Kit | Illumina | (20015964) | Commercial assay or kit |
| Chemical compound, drug | Phos-tag Acrylamide AAL-107 | Alpha Laboratories | (304–93521) | Chemical |
| Chemical compound, drug | Nocodazole | Merck Life Science | (M1404) | Chemical |
| Chemical compound, drug | 4-Nitroquinoline N-oxide | Merck Life Science | (N8141) | Chemical |
| Software, algorithm | LocalMapper software | *Batrakou et al., 2020* | | |
| Software, algorithm | Repliscope software | *Batrakou et al., 2020* | | |

## Growth conditions and yeast strains

Cell growth, arrests, flow cytometry, and yeast protein extracts were performed as previously described (*Zegerman and Diffley, 2010*).

| *S. cerevisiae* strains | All the strains used in this work are derived from W303 (*ade2-1 ura3-1 his3-11,15 trp1-1 leu2-3,112 can1-100, rad5-535*).<br>SLD3 and DBF4 mutants that cannot be phosphorylated by RAD53 (*sld3-38A* and *dbf4-4A*) are abbreviated to *sld3-A dbf4-A* in the main text. |
|---|---|
| yPZ 1223 | MATa SLD3-10his13myc::KanMX bar1Δ::hisG |
| yPZ 1319 | MATa sld3-38A-10his13myc::KanMX bar1Δ::hisG |
| yPZ 1317 | MATa DBF4-13myc::KanMX bar1Δ::hisG |
| yPZ 705 | MATa |
| yPZ 4076 | MATa trp1::P$_{GAL}$-swi4-t::TRP1 |
| yPZ 4137 | MATa dbf4-4A::HIS3 sld3-38A-10his13myc::KanMX trp1::P$_{GAL}$-swi4-t::TRP1 |
| yPZ 917 | MATa dbf4-4A::HIS3 sld3-38A-10his13myc::KanMX |
| yPZ 1523 | MATa sml1Δ::URA3 rad53Δ::LEU2 SLD3-10his13myc::KanMX bar1Δ::hisG |
| yPZ 1522 | MATa sml1Δ::URA3 SLD3-10his13myc::KanMX bar1Δ::hisG |
| yPZ 1198 | MATa cdc28-as1 (F88G) |
| yPZ 1767 | MATa cdc28-as1 (F88G) dbf4-4A::HIS3 sld3-38A-10his13myc::KanMX |
| yPZ 4014 | MATa RRIGA-split LEU2 ChrIII (YCLWdelta5::HphMX-EU2-ARS317-LEU::NFS1), RAD5$^+$ sld3-38A-10his13myc::KanMX dbf4-4A::HIS3 |
| yPZ 4085 | MATa RRIGA-split LEU2 ChrIII (YCLWdelta5::HphMX-EU2-ARS317-LEU::NFS1), RAD5$^+$ trp1::orc6-4A-P$_{GAL1-10}$-Mcm7-2xNLS::TRP1, ura3::P$_{GAL}$-Cdc6-13myc::URA3 |
| yPZ 4089 | MATa RRIGA-split LEU2 ChrIII (YCLWdelta5::HphMX-EU2-ARS317-LEU::NFS1), RAD5$^+$ trp1::orc6-4A-P$_{GAL1-10}$-Mcm7-2xNLS::TRP1, ura3::P$_{GAL}$-Cdc6-13myc::URA3 sld3-38A-10his13myc::KanMX dbf4-4A::HIS3 |
| yPZ 125 | MATa DBF4-13myc::KanMX |
| yPZ 170 | MATa dbf4Δ::TRP1 his3::P$_{DBF4}$-dbf4-19A-13myc::KanMX::HIS3 |
| yPZ 2 | MATa sld3-38A-10his13myc::KanMX |
| yPZ 52 | MATa SLD3-10his13myc::KanMX |
| yPZ 1471 | MATa sml1Δ::URA3 rad53Δ::HphNT1 DBF4-13myc::KanMX bar1Δ::hisG |
| yPZ 1473 | MATa sml1Δ::URA3 DBF4-13myc::KanMX bar1Δ::hisG |
| yPZ 520 | MATa sml1Δ::URA3 SLD3-13myc::KanMX |
| yPZ 89 | MATa sml1Δ::URA3 rad53Δ::LEU2 SLD3-13myc::KanMX |
| yPZ 519 | MATa sml1Δ::URA3 DBF4-13myc::KanMX |
| yPZ 228 | MATa sml1Δ::URA3 rad53Δ::LEU2 DBF4-13myc::KanMX |

## Yeast strain notes

*dbf4-4A* refers to the Rad53 site mutant. It has serine/threonine to alanine mutations at amino acids 518, 521, 526, 528.

*dbf4-19A* refers to the Rad53 site mutant. It has serine/threonine to alanine mutations at amino acids 53, 59, 188, 192, 203, 222, 224, 226, 228, 318, 319, 328, 374, 375, 377, 518, 521, 526, 528.

*sld3-38A* refers to the Rad53 site mutant. It contains serine/threonine to alanine mutations at amino acids 306, 310, 421, 434, 435, 438, 442, 445, 450, 451, 452, 456, 458, 459, 479, 482, 507, 509, 514, 519,521,524, 540, 541, 546, 547, 548, 550, 556, 558, 559, 565, 569, 582, 607, 653 and 654. 539 is mutated to arginine.

*orc6-4A* refers to CDK sites 106, 116, 123, and 146 mutated to alanine. Swi4-t refers to C-terminally truncated protein 1–814.

## Western blot

Western blots were performed as previously described (*Can et al., 2019*). Rad53 was detected with ab104232 (Abcam, dilution 1:5000).

## Replication profiles

Yeast genomic DNA was extracted using the smash and grab method (https://fangman-brewer.genetics.washington.edu/smash-n-grab.html). DNA was sonicated using the Bioruptor Pico sonicator (Diagenode), and the libraries were prepared according to the TruSeq Nano sample preparation guide from Illumina. To generate replication timing profiles, the ratio of uniquely mapped reads in the replicating samples to the non-replicating samples was calculated following *Batrakou et al., 2020*. Replication profiles were generated using ggplot and smoothed using a moving average in R. The values of Trep were taken from OriDB (*Siow et al., 2012*).

## RRIGA assay using split LEU2 marker

Cells were pre-grown under permissive conditions in YPraff overnight 30°C. At $1 \times 10^7$ cells/ml, nocodazole (2 mg/ml in dimethyl sulfoxide) was added to a final concentration of 10 µg/ml. Cells were arrested for 90 min at 30°C (uninduced time point), and then galactose was added to a final concentration of 2% + fresh nocodazole for 3 hr (induced time point).

For each time point, a 0.5 ml sample was spun at 3.2 K for 1 min in benchtop centrifuge; cells were washed with 1 ml sterile water to remove YP, respun, and resuspended in 0.5 ml sterile water. Cells were sonicated briefly to ensure cells are separated and then a serial dilution was made into sterile water as follows: dilution 1: 10 µl cells + 990 µl water = approx $1 \times 10^5$ cells/ml = $1 \times 10^2$ cells/µl; dilution 2: 10 µl dilution 1 + 990 µl water = approx $1 \times 10^3$ cells/ml = 1 cell/µl.

One hundred microlitre of dilution 2 was plated on YPD plates in triplicate for the viability calculation. One hundred microlitre and 10 µl of undiluted cells were plated in triplicate on SC-leu plates to obtain the fraction of viable cells that are Leu$^+$ before and after induction of re-replication. Plates were incubated at 25°C for 48 hr before colonies were counted. To calculate the percentage of cells in the population that were Leu$^+$, the number of Leu$^+$ colonies per ml was divided by the number of viable cells per ml.

## Acknowledgements

We thank members of the Zegerman lab for critical reading of the manuscript. Work in the PZ lab was supported by AICR 10–0908, Wellcome Trust 107056/Z/15/Z, Cancer Research UK C15873/A12700 and Gurdon Institute funding (Cancer Research UK C6946/A14492, Wellcome Trust 092096). Part III undergraduate student DA was supported by the Department of Biochemistry. MS was funded by the BBSRC BB/M011194/1. GC was supported by a Turkish government grant and a Raymond and Beverley Sackler studentship.

## Additional information

### Funding

| Funder | Grant reference number | Author |
| --- | --- | --- |
| Worldwide Cancer Research | AICR 10-0908 | Mark C Johnson<br>Geylani Can<br>Miguel Monteiro Santos<br>Diana Alexander<br>Philip Zegerman |
| Wellcome Trust | 107056/Z/15/Z | Mark C Johnson<br>Geylani Can<br>Miguel Monteiro Santos<br>Diana Alexander<br>Philip Zegerman |
| Biotechnology and Biological Sciences Research Council | BB/M011194/1 | Miguel Monteiro Santos |
| Cancer Research UK | C15873/A12700 | Mark C Johnson<br>Geylani Can<br>Miguel Monteiro Santos<br>Diana Alexander<br>Philip Zegerman |
| Cancer Research UK | C6946/A14492 | Mark C Johnson<br>Philip Zegerman<br>Miguel Monteiro Santos<br>Diana Alexander<br>Geylani Can |
| Wellcome Trust | 092096 | Mark C Johnson<br>Geylani Can<br>Miguel Monteiro Santos<br>Diana Alexander<br>Philip Zegerman |

The funders had no role in study design, data collection and interpretation, or the decision to submit the work for publication.

### Author contributions

Mark C Johnson, Geylani Can, Miguel Monteiro Santos, Diana Alexander, Formal analysis, Methodology; Philip Zegerman, Conceptualization, Formal analysis, Investigation, Methodology, Writing - original draft, Project administration

### Author ORCIDs

Mark C Johnson https://orcid.org/0000-0002-6136-7055
Geylani Can http://orcid.org/0000-0002-1716-7830
Miguel Monteiro Santos http://orcid.org/0000-0002-5594-2682
Diana Alexander http://orcid.org/0000-0002-7785-3170
Philip Zegerman https://orcid.org/0000-0002-5707-1083

### Decision letter and Author response

Decision letter https://doi.org/10.7554/eLife.63589.sa1
Author response https://doi.org/10.7554/eLife.63589.sa2

## Additional files

### Supplementary files

• Transparent reporting form

## Data availability

Sequencing data has been deposited in GEO under the accession code GSE159122 and GSE163571.

The following datasets were generated:

| Author(s) | Year | Dataset title | Dataset URL | Database and Identifier |
|---|---|---|---|---|
| Johnson MC, Can G, Santos MM, Alexander D, Zegerman P | 2020 | Checkpoint inhibition of origin firing prevents inappropriate replication outside of S-phase | http://www.ncbi.nlm.nih.gov/geo/query/acc.cgi?acc=GSE159122 | NCBI Gene Expression Omnibus, GSE159122 |
| Johnson MC, Can G, Santos MM, Alexander D, Zegerman P | 2020 | Checkpoint inhibition of origin firing prevents inappropriate replication In G2/M phase. | http://www.ncbi.nlm.nih.gov/geo/query/acc.cgi?acc=GSE163571 | NCBI Gene Expression Omnibus, GSE163571 |

The following previously published dataset was used:

| Author(s) | Year | Dataset title | Dataset URL | Database and Identifier |
|---|---|---|---|---|
| Morafraile EC, Hanni C, Allen G, Zeisner T, Clarke C, Johnson MC, Santos MM, Carroll L, Minchell NE, Baxter J, Banks P, Lydall D, Zegerman P | 2019 | Genetic screens identify pathways that are important in the absence of checkpoint inhibition of origin firing | http://genesdev.cshlp.org/content/33/21-22/1539/suppl/DC1 | Supplemental_SuppTable3.xlsx, DC1 |

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
