## [Decision Letter]

**Acceptance summary:**

Johnson et al. describe roles for the "S-phase checkpoint" outside of S phase, extending the reach of the factors responsible for monitoring DNA integrity to other parts of the cell cycle. This careful analysis of the general roles of this checkpoint in yeast have obvious parallels for understanding checkpoint functions more generally, and the DNA damage response in higher eukaryotes that are less amenable to the types of analysis performed here.

**Decision letter after peer review:**

Thank you for submitting your article "Checkpoint inhibition of origin firing prevents inappropriate replication outside of S-phase" for consideration by *eLife*. Your article has been reviewed by three peer reviewers, including Tim Formosa as the Reviewing Editor and Reviewer #1, and the evaluation has been overseen by Kevin Struhl as the Senior Editor. The following individuals involved in review of your submission have agreed to reveal their identity: Bruce Stillman (Reviewer #2); Karim Labib (Reviewer #3).

The reviewers have discussed the reviews with one another and the Reviewing Editor has drafted this decision to help you prepare a revised submission.

Summary:

Johnson et al. use the yeast *Saccharomyces cerevisiae* system to show that oxidative damage triggers phosphorylation of DNA replication initiation factors Sld3 and Dbf4 by activated Rad53 kinase in cells arrested in either G1 or G2/M. They also show that regulating the activities of these factors outside of S-phase involves the same phosphorylation target sites that are used within S-phase. Finally, they show that the inappropriate replication that occurs in the presence of damage when the checkpoint is inactivated follows the normal timing program, with early origins activated first. These results show that the same checkpoint controls that monitor S phase progression are also used during other phases of the cell cycle. The effects during G1 provide a direct mechanism for inhibiting initiation, in addition to the previously studied parallel block to S phase entry produced by the effects of Rad53 on CDK activation. The effects in G2 limit re-replication, which was previously shown to be regulated by CDK activity as well. This work takes advantage of the high resolution available in the yeast system to concisely establish a broader role for the S-phase checkpoint system than had been previously assumed, and to cleanly demonstrate how multiple factors contribute to maintaining genomic stability in the presence of DNA damage.

Essential revisions:

1) The authors should submit their DNA sequence copy number data to a public DNA sequence database

2) Introduction: The authors should provide further support for the contention that S-phase checkpoint controls are currently assumed to act only within S phase.

3) Subsection “Rad53-dependent phosphorylation of Sld3 and Dbf4 reduces re-replication in G2 phase” and Figure 3B: The change in flow cytometry profiles seems small; have potential changes in cell size (FSC, SSC) and the variation in mitochondrial DNA associated with cell size been accounted for?

4) Introduction: Parallels to the human DNA damage checkpoints should be included here, especially the role of p21-CIP. In general, more discussion of the parallel contributions of phosphorylation by Rad53 to Sld3-Dbf4 regulation and CDK activity would strengthen the manuscript. The recent results from the Remus laboratory on the effects of Rad53 on the MCM complex would also provide additional context for the Sld3-Dbf4 results described here.

5) Figure 1B upper panel, Figure 1—figure supplement 1B upper panel, Figure 2A upper panel: Sld3 appears as a doublet in untreated G1 cells (0 minute sample). Similarly, in Figure 1E upper panel, the bottom band that is supposed to represent unphosphorylated Sld3 only decreases slightly upon DNA damage. The authors should demonstrate that the lower band is specific to Sld3 and explain these features of the data.

6) Figure 2B, D: The level of the dbf4-A mutant protein appears to be higher in cells compared to the Dbf4 WT protein. As Dbf4 is limiting for initiation and the effects of combining dbf4-A with swi4-t in Figure 4 are rather subtle, the authors should consider the possibility that the effects are due to overexpression of dbf4-A protein rather than to the lack of phosphorylation.

7) Subsection “Rad53-dependent phosphorylation of Sld3 and Dbf4 reduces re-replication in G2 phase” and Figure 3D: The statistical significance of the small change reported with 3 replicates suggests that further repeats would be needed to support the conclusion here and in Figure 4F.

8) The use of dbf4 alleles to establish which phosphorylation sites are important should be clarified. It is not always clear whether the 4A allele (mutations in the sites that are essential in S phase) or the 19A allele (mutations in all mapped phosphorylation sites) are being used in specific experiments. This could affect whether or not the conclusion that the same sites are important in S phase and G1/G2 is valid, so this needs to be clarified and the reasons for using different alleles justified.

---

## [Author Response]

Essential revisions:1) The authors should submit their DNA sequence copy number data to a public DNA sequence database

The GEO accession number is GSE159122.

2) Introduction: The authors should provide further support for the contention that S-phase checkpoint controls are currently assumed to act only within S phase.

Here we make the general point that checkpoints are considered to be tailored to different cell cycle phases. To support this point, we have included the citation (Shaltiel et al., 2015) which states in the Abstract “despite the shared mechanisms of DNA damage detection throughout the cell cycle, the checkpoint and its reversal are precisely tuned to each cell cycle phase”. We think that our analysis of checkpoint inhibition of origin firing throughout the cell cycle shows that for at least this branch of the “S-phase” checkpoint there is no evidence of tailoring to a single phase.

3) Subsection “Rad53-dependent phosphorylation of Sld3 and Dbf4 reduces re-replication in G2 phase” and Figure 3B: The change in flow cytometry profiles seems small; have potential changes in cell size (FSC, SSC) and the variation in mitochondrial DNA associated with cell size been accounted for?

Cell size has been accounted for in the flow cytometer as FSC and SSC of all samples was the same. We decided that the best way to demonstrate genomic re-replication was to perform DNA sequencing of the experiment described in this comment by the reviewers. By over-expressing a subset of pre-RC mutants that are refractory to inhibition by CDK (Mcm7-NLS, Orc6-A, PGal-Cdc6) we observed re-replication mostly on Chromosome III (see new Figure 3—figure supplement 1C), which is in line with previous work (Green et al., 2006). This limited re-replication also underlines why the changes in DNA content by flow cytometry are modest even in the Mcm7-NLS, Orc6-A, PGal-Cdc6 strain. Importantly we observe greater re-replication of chromosome III when Sld3 and Dbf4 cannot be inhibited by Rad53 (new Figure 3B). From this sequencing we also observe that mitochondrial DNA content is equivalent between the strains (new Figure 3—figure supplement 1B). Together with the other data in Figure 3, we are confident that Rad53 can inhibit Sld3 and Dbf4 in G2 phase to limit the rate of re-replication.

4) Introduction: Parallels to the human DNA damage checkpoints should be included here, especially the role of p21-CIP. In general, more discussion of the parallel contributions of phosphorylation by Rad53 to Sld3-Dbf4 regulation and CDK activity would strengthen the manuscript. The recent results from the Remus laboratory on the effects of Rad53 on the MCM complex would also provide additional context for the Sld3-Dbf4 results described here.

We are very grateful for these comments and we have greatly expanded this section of the Introduction and included the points and references suggested. We then circle back to the parallel pathways of CDK and replication factor inhibition in the Discussion.

5) Figure 1B upper panel, Figure 1—figure supplement 1B upper panel, Figure 2A upper panel: Sld3 appears as a doublet in untreated G1 cells (0 minute sample). Similarly, in Figure 1E upper panel, the bottom band that is supposed to represent unphosphorylated Sld3 only decreases slightly upon DNA damage. The authors should demonstrate that the lower band is specific to Sld3 and explain these features of the data.

Sld3 has been shown to be phosphorylated by DDK in G1 arrested cells (Mattarocci et al., 2014) and we have included a sentence (subsection “Sld3 and Dbf4 are phosphorylated by Rad53 outside of S-phase”) to state that we consider it likely that the minor phospho-Sld3 band in the G1 arrested samples is due to DDK phosphorylation.

We are not sure why there is a persistent unphosphorylated fraction of Sld3 in nocodazole arrested cells (e.g. Figure 1E), which does not change after DNA damage. We include a new figure (Figure 1—figure supplement 1C) without tagged Sld3, with and without inhibition of CDK, which clearly identifies these bands as being specific to CDK phosphorylated and unphosphorylated Sld3.

6) Figure 2B, D: The level of the dbf4-A mutant protein appears to be higher in cells compared to the Dbf4 WT protein. As Dbf4 is limiting for initiation and the effects of combining dbf4-A with swi4-t in Figure 4 are rather subtle, the authors should consider the possibility that the effects are due to overexpression of dbf4-A protein rather than to the lack of phosphorylation.

In this study we have used two different mutants of Dbf4; 19A, which lacks all the Rad53 phospho-sites and 4A, which lacks the key Rad53 sites that inhibit Dbf4’s role in origin firing. (Note that the 4A sites are a subset of the 19A sites). Also, in light of point 8 raised by the reviewers, we realise that this is confusing and we did not explicitly state which allele was used in Figure 2, for which we apologise. For all phenotypic assays (including Figure 4) we always use Dbf4-4A, because this allele is fully functional for origin firing and the protein is expressed at wild type levels. Therefore, the expression level of Dbf4-4A is not an issue for Figure 4.

For Figure 2B and D (and only in these figures) we used Dbf4-19A, because it lacks all available Rad53 phospho-sites and therefore gives a more robust loss of phosphorylation by western blot. As pointed out by the reviewers, this 19A allele is more expressed than the wild type protein (and it’s a hypomorph, which may be why cells adapt by over-expressing it). We have rectified these issues as follows:

1) At the first mention of Dbf4 phospho-site mutants (subsection “Sld3 and Dbf4 are phosphorylated by Rad53 outside of S-phase”), we explain the difference between these 2 alleles. From then on, we refer to Dbf4-4A as Dbf4-A and Dbf4-19A as it is.

2) We clearly state that Dbf4-19A was used for Figure 2B and D in the figure, legend and the text.

3) We have included a new figure which shows that Dbf4-4A is expressed at very similar levels to wild type and also shows defects in Rad53 phosphorylation in G1 phase (new Figure 2—figure supplement 1).

7) Subsection “Rad53-dependent phosphorylation of Sld3 and Dbf4 reduces re-replication in G2 phase” and Figure 3D: The statistical significance of the small change reported with 3 replicates suggests that further repeats would be needed to support the conclusion here and in Figure 4F.

The statistical significance of these differences is less than 0.05, so the data supports our conclusions. More repeats would indeed increase the significance further.

8) The use of dbf4 alleles to establish which phosphorylation sites are important should be clarified. It is not always clear whether the 4A allele (mutations in the sites that are essential in S phase) or the 19A allele (mutations in all mapped phosphorylation sites) are being used in specific experiments. This could affect whether or not the conclusion that the same sites are important in S phase and G1/G2 is valid, so this needs to be clarified and the reasons for using different alleles justified.

See point 6.